# Functional Roles of the Seagrass (*Zostera marina*) Holobiont Change with Plant Development

**DOI:** 10.3390/plants14111584

**Published:** 2025-05-23

**Authors:** Sam Gorvel, Bettina Walter, Joe D. Taylor, Richard K. F. Unsworth

**Affiliations:** 1Seagrass Ecosystem Research Group, Swansea University, Singleton Park, Swansea SA2 8PP, UK; sgorvel13@gmail.com (S.G.); bettiwalter@web.de (B.W.); 2Project Seagrass, Unit 1 Garth Drive, Brackla Industrial Estate, Bridgend CF31 2AQ, UK; 3UK Centre for Ecology & Hydrology, Maclean Building, Benson Lane, Crowmarsh Gifford, Wallingford OX10 8BB, UK; joetay@ceh.ac.uk

**Keywords:** eelgrass, microbiome, restoration, marine, microbial

## Abstract

Seagrass meadows play a critical role in biogeochemical cycling, especially in nitrogen and sulphur processes, driven by their associated microbiome. This study provides a novel functional analysis of microbial communities in seagrass (*Zostera marina*) rhizosphere and endosphere, comparing seedlings and mature plants. While nitrogen-fixing bacteria are more abundant in seedlings, mature plants exhibit greater microbial diversity and stability. Sediment samples show higher microbial diversity than roots, suggesting distinct niche environments in seagrass roots. Key microbial taxa (sulphur-oxidizing and nitrogen-cycling bacteria) were observed across developmental stages, with rapid establishment in seedlings aiding survival in sulphide-rich, anoxic sediments. Chromatiales, which oxidize sulphur, are hypothesized to support juvenile plant growth by mitigating sulphide toxicity, a key stressor in early development. Additionally, sulfate-reducing bacteria (SRB), though potentially harmful due to H_2_S production, may also aid in nitrogen fixation by producing ammonium. The study underscores the dynamic relationship between seagrass and its microbiome, especially the differences in microbial community structure and function between juvenile and mature plants. The study emphasizes the need for a deeper understanding of microbial roles within the seagrass holobiont to aid with Blue Carbon stores and to improve restoration success, particularly for juvenile plants struggling to establish effective microbiomes.

## 1. Introduction

The supportive functions of seagrass such as habitat, shelter, and trophic provide extensive support for biodiversity [1,2]. In addition, seagrass meadows also provide regulatory services such as carbon storage, nitrogen cycling and coastal protection. Literature around these services commonly suggests seagrass as their sole provider, however, it is far more likely that the responsibility for such services instead lies with its overarching holobiont assemblage [3].

Seagrass ameliorates surrounding sediment via oxygen expulsion from root pores [4,5] allowing the formation of distinct local microenvironments where oxygen is no longer limiting [4]. Negating oxygen limitation is key to seagrass success, as important activity by chemolithotrophic and aerobic detoxifying microorganisms are then thought to prevent die-offs through prevention of sulphide accumulation [6].

Similarly, evidence suggests *Zostera*-mediated root exudates (formed principally of labile organic matter) attract a diverse range of heterotrophic bacteria such as Gamma and Deltaproteobacteria [4,6]. These heterotrophic bacteria then negatively influence oxygen balance, enhancing the cycling of carbon. Seagrasses also live across a range of nutrient environments from offshore and island calcareous sands low in nutrients to muddy organic rich lagoons awash with nutrients. In many parts of the world such environments have become increasingly eutrophic [7], however seagrasses are poorly adapted to uptake of nitrate instead requiring ammonia [8]. Cycling of this nitrate and making it more bio-available requires microorganisms, with seagrass-associated micro-organisms thought to be involved in the nitrification, denitrification and nitrogen-fixing processes [9].

*Zostera marina* root systems assemble two distinct populations (i.e., *microbiome*) from existing sediment-microbe reservoirs: rhizosphere (surrounding roots) and endosphere (within roots) [10]. Although there is generally a limited literature into root-endosphere community diversity [11] growing evidence suggests high proportions of sulfate-reducing bacteria (SRB) colonize root tissues [12]. There is increasing evidence that these seagrass associated microbial communities change with respect to environmental differences [13] and may play a key role in improving the resilience of the plants to disease [14].

There are many knowledge gaps within how seagrass root-associated communities develop, with many studies being temporally isolated with a limited understanding of seagrass life history stages. Local environmental differences have been found to create rapid shifts in associated microbial community composition [13,15] with small scale spatial differences known to exist, suggesting host-selection of some functional taxa [15]. However, no indication is made as to whether these communities persist over time within seagrass meadows, or if these microbes are involved with *Z. marina* development. These bacterial assemblages interact directly with *Z. marina* across all stages of their lifecycle. They are evident in seed germination, all subsequent successional development stages, and adult plants, whilst they have also been observed to play a role in the degradation of detrital plant matter [10,16,17,18].

Knowledge of the role of microbial communities in seagrass meadows has direct links to the emerging field of Blue Carbon (organic carbon stored in the world’s coastal and marine ecosystems) [19], as seagrasses have the capacity to act as both sinks and sources of Greenhouse Gas Emissions. Microbes that are involved with or influence the breakdown of organic matter, production of methane, and conversion of nitrate to nitrous oxide have potential to alter these roles [6,20]. Understanding of the drivers behind the productivity of these organisms has potential to influence how these systems are managed into the future.

The emerging view of the seagrass holobiont and the factors that influence it are also of increasing importance given the growth of activity in seagrass restoration [21]. Failure in seagrass restoration is widespread, and in many examples there remains limited understanding as to why a project failed, and as a result many practitioners are looking at alternative explanations such as the role of the microbiome [22]. In other marine ecological fields, such as coral reef restoration and aquaculture, there is increasing hope that probiotics can be developed that can improve growth and survival of corals, particularly in the context of extreme environmental conditions [23,24]. In seagrasses, if probiotics could be developed or conditions altered to facilitate improved colonisation by the most appropriate microbes then there is hope that some of the bottlenecks for seagrass restoration could be overcome [21]. These hypothetical innovations first necessitate a better understanding as to the key functional communities and species of microbes present in seagrass at different stages of their development and in natural environments. It is also important that this understanding is developed based on an understanding of real-world field experiments as many of the experiments and studies conducted on the seagrass microbiome have bee conducted in controlled laboratory conditions.

Given the limited understanding of the seagrass microbiome with respect to different stages of life history and the recognition that filling gaps in this knowledge may help improve our understanding of Blue Carbon as well as the recovery and restoration of these systems, the following experiment sought to examine the seagrass microbiome at different stages of plant development. This study aimed to examine whether such a community exists within young seedlings and whether the functional role of these communities develops and changes with the age of the plant.

## 2. Materials and Methods

### 2.1. Study Sites

Samples of *Zostera marina* and adjacent sediments associated with the plant were collected from two seagrass planting (restoration) sites and two natural seagrass meadow sites located in Milford Haven Waterway, South Wales (Figure 1). All sites within the Milford Haven Waterway observe an annual temperature range of 8–17 °C peaking in August [25]. The temperate seagrass *Z. marina* covers large areas of mostly subtidal soft sediment substrate while experiencing high water renewal with a maximum tidal range of 7.68 m. Gelliswick Bay has predominantly fine-very fine sand overlying silt/clay [25]. All sites associated with the Dale tidal lagoon (treatments, restoration, Dale patch) (Figure 1) are dominated by fine to very fine sandy sediments at depths ranging 1–3 m [22]. Existing *Z. marina* meadows found in Gelliswick have been dated back over 100 years [26] suggesting prime conditions for seagrass growth. The Dale tidal lagoon was at the time of the study largely devoid of any seagrass except for the presence of an isolated patch of dense *Z. marina* (approx. 5 m^2^) surrounded by bare sediment. Prospecting studies [22] confirmed reports from as early as the mid-1950s [26] as well as ratifying model outputs that predicted *Z. marina* presence in this area [27].

Seagrass planting sites (restoration trials) located no further than 50 m from the Dale patch observe very similar conditions. The restoration sites represents a random scattering of hessian bags loaded with *Z. marina* seeds and *Z. marina*-derived detrital inoculant [22]. Deployment of the restoration materials (seeds) occurred during the summer of 2017. Deployment was performed by wading out on low tides with lines that were weighted down and buoyed (described in [28]).

### 2.2. Sample Collection

Seagrass plants were sampled randomly from the restoration trial within their respective age ranges. Tissue from *Z. marina* roots, rhizomes and associated sediments were collected in water during March, May and July 2019, with collectors wearing nitrile gloves and clear protocols ensuring no cross-contamination between samples. This period was used in order to collect plants over a range of ages but also the critical periods of seedling emergence known at the site [22]. The base of the plant was accessed by gently loosening sediment until whole root-rhizome structure was removed. Plants were then immediately rinsed with filter sterilised saline solution of 35 ppt to remove remaining sediment from the plants. Sediment samples were collected from random locations adjacent to the sampled seagrass plants. Sediment samples were collected using 5 mL syringes modified with the ends cut off to form mini-corers (Surface area 1 cm^2^), 5 cm depth. Samples were then immediately frozen in dry-ice (−60 °C) and transferred to the freezer −80 °C until extraction.

### 2.3. DNA Extraction, PCR Amplification and High-Throughput Sequencing

DNA extractions of root and rhizome tissue samples and associated sediment samples were extracted using a DNeasy PowerSoil kit (Qiagen, Venlo, The Netherlands) following the manufacturer’s protocol. Root and rhizome samples were thawed from −80 °C storage tubes, transferred to sterile petri dishes and cut into equal ~0.5 cm segments using a blade sterilized under flame and cleaned with 70% ethanol. Tissue samples were then directly loaded into several PowerBead tubes with 3–5 roots or rhizome segments from each plant. Sediment samples were voided from syringes and separated at a fixed depth of 5 cm. Sediment was then homogenized using a Vortex-Genie 2.2 (Scientific Industries, New York) briefly. PowerBead tubes were then loaded with 0.4 g of homogenized sediment.

DNA extracts were then sent to Integrated Microbiome Resource (Halifax, NS, Canada) for PCR and sequencing. The bacterial community was amplified using PCR primers for the 16S rRNA V4 amplicon fragments using high-fidelity Phusion polymerase and fusion primers containing Illumina sequencing adapters and the primer sequences Forward 515FB 5′-GTGYCAGCMGCCGCGGTAA-3′, Reverse 806RB 5′-GGACTACNVGGGTWTCTAAT-3′. Amplicon fragments were PCR-amplified from the DNA in duplicate using separate template dilutions (1:1 & 1:10) using the high-fidelity Phusion Plus polymerase. PCR products are verified visually by running on a high-throughput Hamilton Nimbus Select robot using Coastal Genomics Analytical Gels. The PCR reactions from the same samples are pooled in one plate, then cleaned-up and normalized using the high-throughput Charm Biotech Just-a-Plate 96-well Normalization Kit. Libraries were then sequenced on an Illumina MiSeq using 300 + 300 bp paired-end V3 chemistry (Illumina, Inc., San Diego, CA, USA).

### 2.4. Sequence Processing

Sequences were analysed using a combination of Quantative Insights into Microbial Ecology (QIIME) v 1.9.0 [29] and USEARCH v11 [30]. Chimeras were identified using USEARCH v11 and were filtered out. Remaining sequences were grouped into zOTUs (zero-radius Operational Taxonomic Units) using UNOISE3 (Edgar, 2010) [11]. Taxonomy was assigned using assign_taxanomy.py QIIME script against Silva 132 16S database [31] using USEARCH. Archaeal, mitochondrial, chloroplast-derived sequences were removed from the data using filter_taxa_from_otu_table.py script, leaving only bacterial zOTUs. After filtering, root and sediment samples retained high numbers of bacterial reads and zOTU tables were rarefied at 5000 reads for comparison between these two fractions. Rhizome samples contained fewer reads with high levels of chloroplast and mitochondrial reads. For statistical comparisons of data with rhizomes, a lower rarefaction depth of 1000 was used.

### 2.5. Data Analysis

All the summary data are presented as means ± sample standard deviations. To understand the dominant functional groups present within these bacterial communities we only analysed those taxa found to contribute a greater than 0.5% of the total data set. From the initial list of 204 taxa at the Order and Class level (including unclassified taxa) this left 34 taxa (see Appendix A). Those taxa were classified into functional groups according to Table 1 to enable further analysis, this followed information contained within the Prokaryote reference collection by Rosenberg, et al. [32], the FAPROTAX database [33] and Bergey’s Manual of Systematic Bacteriology by Garrity, et al. [34]. Data analysis and visualization was performed in MiniTab v22 and Primer v7 [35].

Univariate analysis of zOTUs of the functional taxonomic groupings was conducted using permutational analysis of variance (PERMANOVA+) [36] using a Euclidean resemblance matrix was employed to analyse the differences in relative abundance between plant age (adult vs. juvenile) and structure (roots vs. sediment). Community composition was analysed on square-root transformed data using a Bray-Curtis similarity resemblance measure and visualised using non-metric multidimensional scaling ordination (nMDS). Sequence data has been uploaded to the European Nucleotide Archive under Bioproject Accession number: PRJEB81660.

## 3. Results

A total of 29 samples were successfully sequenced, with 7 samples from Juvenile roots, 6 samples from adult roots and 16 from sediments. The total number of zOTUs in the total dataset was 8089 in a total of 725,762 sequences. After rarefying the data set to 1000 sequences per sample, a total of 5647 zOTUs remained, 200 different taxa with certain taxonomy at the Order and family level were characterised. Thirty-four of these taxa contributed more than 0.5% to the total dataset and were subsequently used in further analysis. The six most abundant of these taxa were Desulfobacterales, Bacteroidales, Chromatiales, Pirellulales, and Clostridiales (see Figure 2). These six taxa comprised 46% of the total dataset.

Of these abundant taxa only the relative abundance of Clostridiales and Flavobacteriales were significantly effected by plant age whereas all had significantly different relative abundance between the root and the sediment (Table 2 and Figure 2). Chromatiales, Clostridiales and Bacteroidales had higher relative abundance on roots, where as Pirelluales had higher relative abundance in the sediment (Table 2). In the context of the seagrass microbiome, and based on a review of literature, three of these are organisms known to focus their nutrition on the processing of fatty acids and sugars (Flavobacteriales, Pirellulales, and Clostridiales), whilst two are involved with sulphur cycling (Desulfobacterales generally reduce sulphur and Chromatiales generally oxidise sulphur). Bacteroidales are thought to play roles in nutrient uptake and host fitness. The abundant taxa were classified into ten functional groups (see Appendix A), and the total relative abundance of each functional group calculated for each sample. This finds that taxa involved with the breakdown of complex sugars and fatty acids are the most abundant (27.7 ± 5.7%) followed by Sulfate Reducing Bacteria (SRB) (13.4 ± 3.0%) and Sulphur Oxidising Bacteria (8.6± 12) (Figure 3).

Sediment samples were found to have significantly higher zOTU richness (margalefs) (*p* = <0.001), Evenness (Pielou’s) (*p* = <0.001), Shannon (*p* = <0.001) and Simpsons diversity (*p* = <0.001) than root samples (Table 2 and Figure 4). There were some significant interactions though as evenness in the juvenile root samples was high variable and limited difference between adult and juvenile sediment for simpsons diversity and richness. There was also a significant effect of age, with juvenile plants for all these diversity metrics having a lower value than adult plants (*p* = <0.001). The community assemblage of the bacterial communities was also significantly affected by age and structure (see Table 2). The nMDS show how the sediment samples are all tightly clustered into one similar community irrespective of age (Figure 5), but the root samples are highly variable relative to the sediment samples and with respect to their age. Imposing 70% similarity clusters onto the associated nMDS show that these differences are most likely influenced by a range of outliers as some of the juvenile root samples are in separate 70% similarity clusters.

Within root communities some key functional groups were significantly more abundant than in sediment environments, specifically the nitrifiers, and those involved with host health, sulphur oxidation and cellulose breakdown. This was also the case with those bacteria involved with the breakdown of sugars and amino acids (Figure 6 and Figure 7). Nitrogen fixers, those involved with organic matter breakdown and those that produce secondary metabolites all had significantly higher relative abundance in the sediment environments relative to the roots (Table 2 and Figure 6). Sulfate reducing bacteria (SRB) and microbes with mixed functional did not significantly differ with respect to root vs. sediment environments (Table 2) in abundance from root to sediment environments.

Age of the plant did not significantly affect the relative abundance of many of the functional groups. Nitrogen fixing bacteria were significantly more abundant in juvenile plants whilst those involved with cellulose breakdown were significantly more abundant in adult plants. This significant increase in nitrogen fixing bacteria interacted with respect to the root vs. sediment environments (See Figure 6). There was also a significant effect of age on the zOTUs of those taxa involved with the breakdown of organic matter.

## 4. Discussion

Seagrass meadows have significant functional roles in biogeochemical cycling, particularly in the context of nitrogen and sulphur cycling, as well as in the accumulation and breakdown of carbon [37]. The microbiome associated with seagrass plays a significant part in these processes but remains poorly understood [6,38]. In the present study, we provide a novel functional breakdown of the microbial communities associated with the seagrass rhizosphere and endosphere relative to changes in plant development. Whilst differences with respect to plant development are limited to a higher relative abundance of nitrogen Fixing bacteria in seedlings and a lower abundance of those involved with organic matter breakdown, there exist large differences in the diversity of the taxa present within the microbiome of seedlings in comparison to mature plants. We also record the relative abundance and composition of microbes to be far more variable within young plants relative to adult ones. In line with previous literature, greater microbial diversity in the present study is observed from sediment samples (relative to roots) [39,40], highlighting how different niche environments likely exist within these regions near to seagrass roots [41].

Although the juvenile plants (seedlings) observed here were only a few months old, their microbial communities of key functional taxa (e.g., Chromatiales generally oxidise sulphur) was no different to that of adult plants but much more abundant than within the surrounding sediment. We hypothesise that these assemblages develop rapidly on the roots to aid with their capacity to deal with the harsh anoxic sulphide environment whilst enabling them to overcome insufficient resource availability ammonium [42].

Seagrass root systems are typically found anchoring in sediments with anoxic and sulphuric conditions [43]; this environment of harsh redox gradients necessitates that seagrasses manipulate their root environment via exudation [44] and radial oxygen loss (ROL) [45], imposing selection pressures on microbial communities within surrounding sediments [15].

The abundance of Sulfate Reducing Bacteria (SRB) observed in the present study within the roots and sediments of seedlings and mature plants highlights the harsh environment for seagrass plants to develop and survive, as the SRB potentially produce an abundance of toxic H_2_S. Such an abundance of microbes involved with suphur cycling have previously been argued to be indicative of more stressed seagrass plants [46]. Seawater contains an abundance of sulphur, leading to the tendency for anoxic conditions to drive the production of hydrogen sulphide aided by an abundance of Sulfate Reducing Bacteria (SRB). The ROL from seagrass roots has been shown to drive the balance of microbes within the rhizosphere, leading to lower Hydrogen Sulphide concentrations and, therefore, lower relative abundance of Sulfate-reducing bacteria (SRB). The toxicity of sulphide to seagrass makes this microbial role critical in ensuring life in harsh environments is viable. Within individual young seedlings, the relative production of ROL is likely lower than that of mature plants, essentially due to lower photosynthetic production [44,47]. It is for this reason that we hypothesise that the rapid accumulation of species of Chromatiales is essential for juvenile plants to survive.

This abundance highlights the harsh environment to which a young seedling must endure to develop into a mature plant. This abundance of SRB in the seedling roots might also have other benefits as many SRBs are capable of di-nitrogen (N_2_) fixation, creating ammonium. The toxicity of the H_2_S may therefore be offset by the production of a stable supply of nutrients. This aligns with the presence of an increased relative abundance of targeted N_2_ fixing organisms (Rhizobiales) on the roots of young plants relative to mature ones. Recent evidence has found that in other species of seagrass have found N2-fixing symbionts explicitly inside seagrass root tissue where a symbiotic relationship is hypothesized to exist [48] that provides the plant with ammonia and amino acids to its host in exchange for sugars. The recording on N fixing bacteria may suggest the presence of similar relationships to those recorded in Posidonia.

The presence of ammonium in the rhizosphere from fixation firstly creates a source of nitrogen for the plants, this is critical as we know that although seagrass meadows can be awash with excess nutrient from pollution, we also know that seedlings and many young plants can be nutrient limited due to low pore water nutrient availability [28]. This ammonium also creates a substrate for an abundance of nitrifying bacteria (Nitrosococcales) which is also then potentially broken down into N_2_O via denitrification from species of Gammaproteobacteria and Rhodobacterales (that have mixed functional roles but have abundant taxa that are involved with denitrification). An abundance of denitrifying bacteria within these root and sediment environments creates potential for nitrous oxide production to be significant. Seagrasses within the experimental site are enriched with excess nitrogen as a result of highly elevated nitrate concentrations [23] and values well above 1 mg/L have been observed at the experimental sites [31]. Although data on nitrous oxide emissions from seagrass is weak, knowledge from mangroves and other systems indicates that elevated nitrate concentrations lead to such emissions.

In contrast to the high potential for the production of nitrous oxides from these seagrass meadows, we also find limited evidence of an abundance of microorganisms likely to produce methane (although a targeted analysis has not been conducted), another possible greenhouse gas emission from seagrass [34]. We also record the presence of a range of taxa thought to be involved with influencing host health. The Bacteroidales are a key taxon in this respect and were abundant on the seagrass. Root. Although the actual function of these taxa is poorly understood mechanistically, evidence from studies on crops and other terrestrial plants shows their presence commonly correlates with increased growth and reproduction [47]. Key roles in P and N cycling might be the pathways to this role [47].

The microbial assemblages around the roots and sediments were dominated throughout by taxa commonly thought to be involved with the breakdown of complex sugars and amino acids, the most abundant of these being the Flavobacteriales, Pirellulales and Clostridiales. This is not surprising given the high levels of sucrose sugars commonly observed in seagrass-associated sediments as a result of root exudation into the rhizosphere [36]. It is not clear whether this process has a positive or negative influence on the plant. The relative abundance of this functional group didn’t alter significantly with respect to the plant’s age or between root and sediment (however abundance was more variable within juvenile roots), suggesting that the whole wider rhizosphere and endosphere region around the plant is awash with these sugars. Heterotrophic microbial degradation of sugars and amino acids is likely to lead to an increased pull down of Oxygen, although not shown to be differing between plant ages, such reduction in oxygen may have a consequential role of causing the reduction of elemental sulphur to hydrogen sulphide, exacerbating the toxicity of what is already a harsh sedimentary environment.

We observed a far more variable microbial assemblage in terms of taxa abundance and competition in the juvenile seagrass relative to adult systems. This may simply be because older, more mature meadows establish a more integrated and active microbiome over time, as seen in their terrestrial counterparts [45]. The higher numbers of observed bacterial OTUs isolated in adult samples may suggest an increased metabolic requirement of established seagrass meadows, as the formation of more complex micronutrients is known to progress with plant age [3]. Adult plants also had higher relative abundance of microbes involved with breakdown of cellulose, likely the result of the established turnover of plant material within a mature meadow rather than individual seedlings.

The microbiome associated to the roots of both juvenile and adult plants is consistently different to the assemblage present in the sediment rhizosphere, highlighting the niche separation between these environments, but also the likely presence of a holobiont community. Nitrifying bacteria, microbes involved with host health and sulphur oxidizing bacteria all had relative abundance elevated at the roots relative to those in the sediment. This classification we use here approximates the functions of these organisms that is an oversimplification of the abundant species present within each taxonomic grouping. To fully understand these groupings and the exact roles being undertaken (e.g., in terms of host health) requires a far deeper look at the individual grouping as well as more of a metagenomics, metatranscriptomics or metabolomics pipeline approach for the mechanistic interrogation of the microbiome [42].

In summary, this study highlights the dynamic relationship between seagrass (*Zostera marina*) and its microbiome, revealing significant differences in microbial communities and functions across plant developmental stages. Juvenile seagrass roots showed higher abundances of nitrogen-fixing bacteria, aiding in nutrient availability and reducing anoxic stress. In contrast, mature plants exhibited more stable and diverse microbial communities. Sediment samples consistently had higher microbial diversity than root samples, indicating distinct niche environments within seagrass meadows. These findings highlight the crucial role of the seagrass holobiont in nitrogen and sulphur cycling, with implications for seagrass restoration and Blue Carbon strategies. Although the present study teases out the exact microbial species present on juvenile seagrass plants it highlights that these communities are very different in composition to those on established adult plants, whether this is an adaptive response to life as seedling in a harsh environment or a desperate attempt to play microbial ‘catch’up’ to develop an effective microbiome remains to be seen. What we do know is that the re-establishment of seagrass populations through various types of restoration activity is riddled with failures and that processes of scale help plants overcome feedbacks in the system [21]. Better understanding of the specific functions of microbial taxa and the seagrass holobiont in overcoming these feedbacks might facilitate employing advanced techniques to improve seagrass ecosystem conservation and restoration.

## Figures and Tables

**Figure 1 plants-14-01584-f001:**
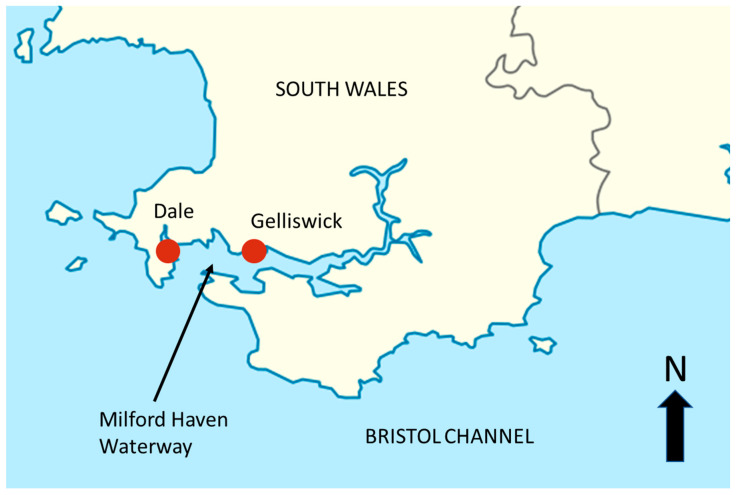
Location of sampling sites in Milford Haven Waterway, South Wales, UK (Dale: 51.7048°, −5.16081° and Gelliswick: 51.707506°, −5.060593°).

**Figure 2 plants-14-01584-f002:**
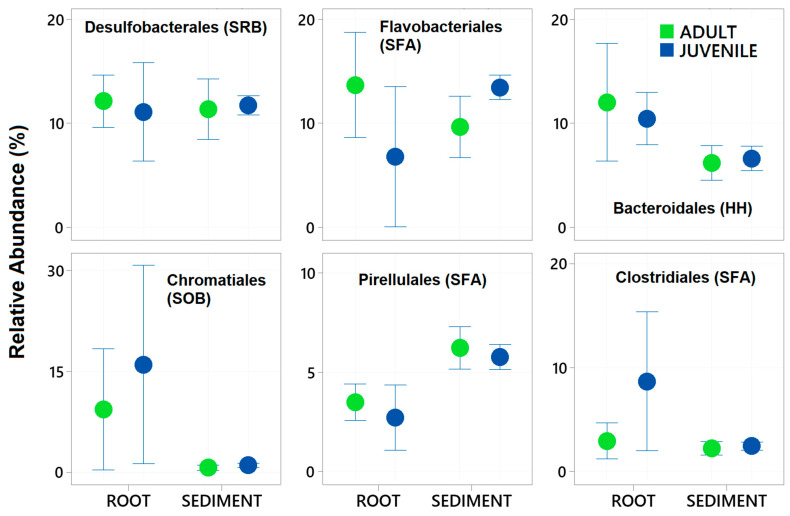
Mean relative abundance ± standard deviation of the six most abundant Bacteria taxa (Order level) in 16S rRNA gene libraries generated from samples of roots and adjacent sediments of juvenile (seedlings) and adult plants of the seagrass *Zostera marina* planted in Dale in the Milford Haven Waterway in West Wales. Green circles are adult plants and blue circles are juvenile plants (seedlings).

**Figure 3 plants-14-01584-f003:**
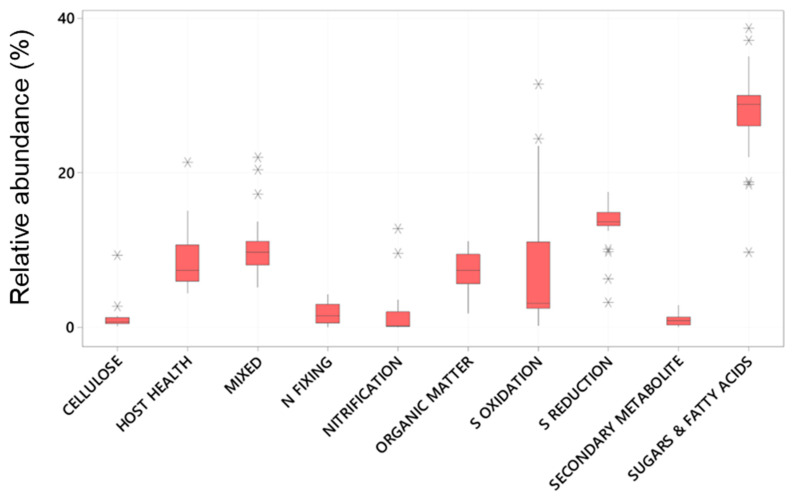
Boxplots of the relative abundance (%) of Bacterial taxa (zOTUs) assigned to functional microbial groupings present on the roots and adjacent sediments of juvenile (seedlings) and adult plants of the seagrass *Zostera marina* planted in Dale in the Milford Haven Waterway in West Wales.

**Figure 4 plants-14-01584-f004:**
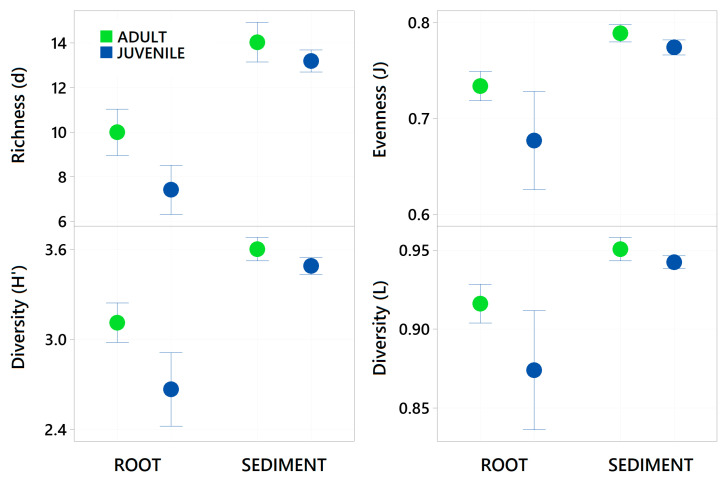
Mean Taxa Richness (margelefs d), Evenness (Pielou’s j), Shannon diversity (H’) and Simpsons Diversity (L) ± standard deviation of all bacterial taxa present (zOTUs) on the roots and adjacent sediments of juvenile (seedlings) and adult plants of the seagrass *Zostera marina* planted in Dale in the Milford Haven Waterway in West Wales. Green circles are adult plants and blue circles are juvenile plants (seedlings), data was normalised by rarefaction to 1000 sequences per sample.

**Figure 5 plants-14-01584-f005:**
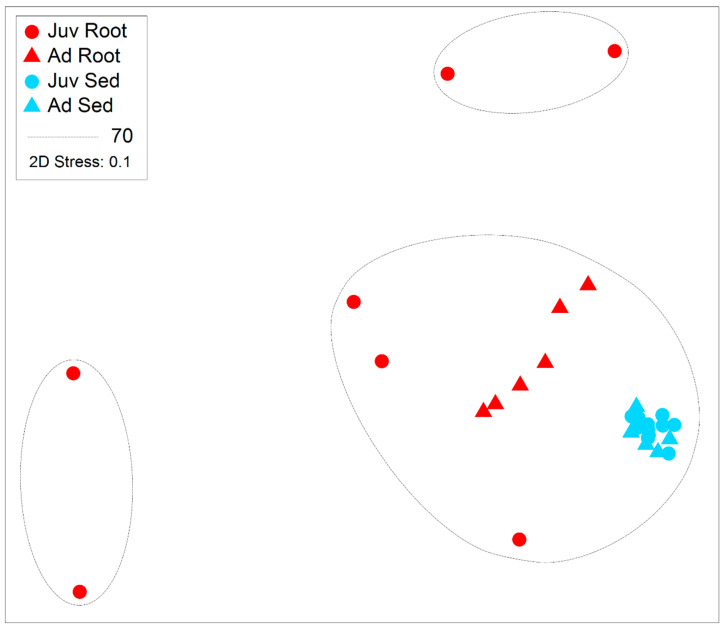
Non-metric multidimensional scaling plot with super-imposed similarity 70% clusters following Bray-Curtis similarity of abundant microbial taxa (zOTUs) present on the roots and adjacent sediments of juvenile (seedlings) and adult plants of the seagrass *Zostera marina* planted in Dale in the Milford Haven Waterway in West Wales, data was normalised by rarefaction to 1000 sequences per sample.

**Figure 6 plants-14-01584-f006:**
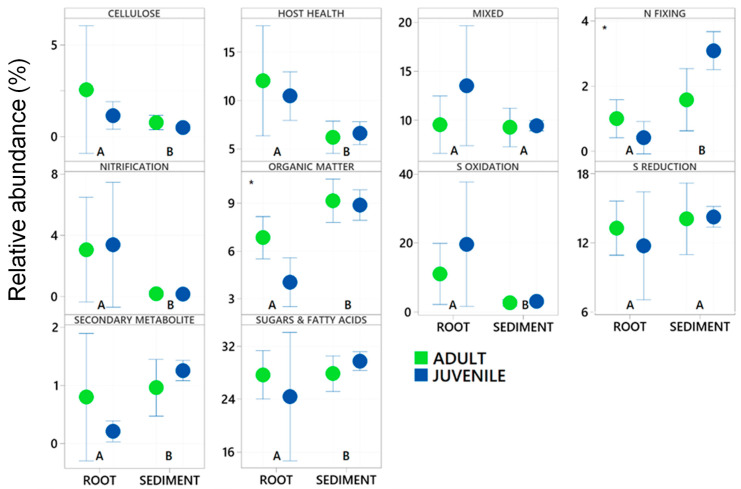
Relative abundance of each of the ten assigned functional microbial groupings to zOTUs within 16S rRNA gene libraries generated from roots and adjacent sediments of juvenile (seedlings) and adult plants of the seagrass *Zostera marina* planted in Dale in the Milford Haven Waterway in West Wales. Significant differences between root and sediment are shown by letters (A & B) whereas significant differences between age classes are shown with a *.

**Figure 7 plants-14-01584-f007:**
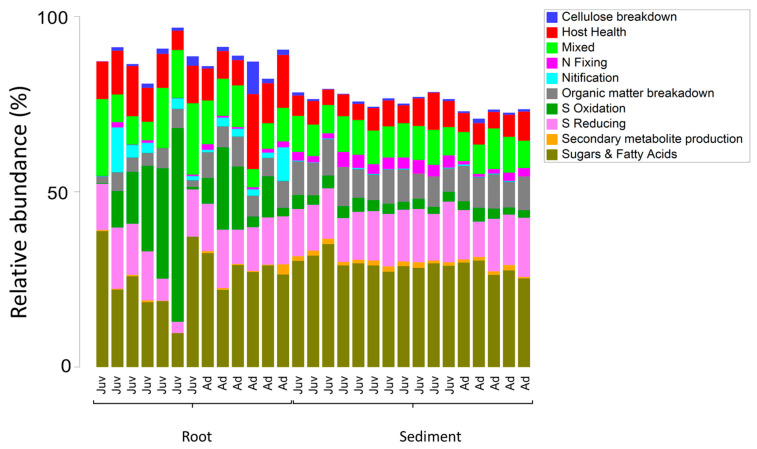
Relative abundance of dominant functional groups of Bacteria in 16S rRNA gene libraries generated from twenty-nine samples collected on the roots and adjacent sediments of juvenile (seedlings) and adult plants of the seagrass *Zostera marina* planted in Dale in the Milford Haven Waterway in West Wales.

**Table 1 plants-14-01584-t001:** Results from two-way PERMANOVA of abundant functional bacterial communities (zOTUs) on the roots and adjacent sediments of juvenile (seedlings) and adult plants of the seagrass *Zostera marina* planted in Dale in the Milford Haven Waterway in West Wales. Analysis of the assemblage is a multivariate analysis whereas the analyses for all the other functional groups and metrics are univariate.

	DoF	Pseudo-F	Age	Pseudo-F	Structure	Pseudo-F	Interaction
Assemblage	1, 1, 25	3.524	0.001	11.14	0.001	2.689	0.001
Richness (d)	1, 1, 25	22.444	0.001	163.05	0.001	7.768	0.011
Evenness (J)	1, 1, 25	10.001	0.004	43.485	0.001	3.703	0.067
Shannon (H’)	1, 1, 25	20.976	0.001	110.22	0.001	8.350	0.01
Simpsons (L)	1, 1, 25	9.286	0.002	37.857	0.001	4.337	0.033
Nitrogen Fixing	1, 1, 25	2.5285	0.033	10.23	0.001	4.186	0.003
Nitrification	1, 1, 25	0.88636	0.413	18.025	0.001	0.244	0.813
S Oxidation (SOB)	1, 1, 25	0.789	0.414	10.08	0.004	1.059	0.314
S Reducing (SRB)	1, 1, 25	0.742	0.419	2.168	0.147	1.087	0.333
Host Health (HH)	1, 1, 25	0.052012	0.913	20.441	0.001	0.40844	0.577
Secondary Metabolites	1, 1, 25	2.2073	0.124	16.467	0.002	5.3014	0.015
Cellulose	1, 1, 25	3.258	0.05	6.625	0.005	0.182	0.869
Mixed	1, 1, 25	1.458	0.245	1.193	0.281	1.046	0.318
Organic	1, 1, 25	8.317	0.006	27.473	0.001	6.603	0.01
Sugars (SFA)	1, 1, 25	1.013	0.421	13.306	0.001	1.786	0.153
Desulfobacterales (SRB)	1, 1, 25	1.751	0.184	14.508	0.001	3.287	0.029
Flavobacteriales (SFA)	1, 1, 25	3.993	0.045	5.009	0.03	17.203	0.002
Bacteroidales (HH)	1, 1, 25	0.052	0.885	20.441	0.001	0.408	0.549
Chromatiales (SOB)	1, 1, 25	0.70781	0.55	14.28	0.001	1.298	0.261
Pirellulales (SFA)	1, 1, 25	3.0534	0.104	30.65	0.001	1.6787	0.225
Clostridiales (SFA)	1, 1, 25	6.8404	0.01	10.278	0.003	4.7801	0.031

**Table 2 plants-14-01584-t002:** Environmental characteristics of the seagrass planting site in Dale, all data taken from Unsworth et al. 2019 and 2022 [22,28].

Parameter	Value
Site depth	0.1 m (below chart datum)
Tidal range	7.7 m
Salinity range	33.5‰
Temperature range	8–17 °C
Porewater nitrogen	280 ± 15 µmol.L^−1^ TON
Seagrass leaf nitrogen	3.5% gDW^−1^
Seagrass phosphorus	0.43% gDW^−1^
Porewater Phosphate	30 ± 0.1 µmol.L^−1^
Porewater Ammonium	25 ± 0.2 µmol.L^−1^

## Data Availability

The original contributions presented in this study are included in the article. Further inquiries can be directed to the corresponding author.

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
