# Peer review of "Functional Roles of the Seagrass (*Zostera marina*) Holobiont Change with Plant Development"

_plants, 2025, doi:10.3390/plants14111584_

Round 1

Reviewer 1 Report

Comments and Suggestions for Authors

This manuscript systematically investigated the functional differences of rhizosphere and endophytic microbial communities in seagrass (Zostera Marina) at different developmental stages (seedlings and adults), revealing the dynamic role of microbial communities in sulfur environment adaptation, nitrogen sulfur cycling, and blue carbon storage. The research results provide a theoretical basis for microbial community regulation in seagrass restoration, with clear ecological significance and application potential. Overall, this manuscript merits publication after major revision. The specific comments are presented below.

  1. The scope of the title of this manuscript is a bit broad, it is suggested to add Zostera marina to the title.
  2. Abstract, Please confirm if “seeding” and “judiciary” have the same meaning and if they need to be expressed uniformly.
  3. Lines 41-42, Generally speaking, only the species names of microorganisms need to be italicized, while other names such as phyla, classes, orders, and families do not need to be italicized. Therefore, the words "Gamma and Deltaproteobacteria" in this sentence do not need to be italicized.
  4. Line 51, Z. marina appears for the first time in the main text and requires the full name "Zostera marina" (italicized), while "Zostera Marina" appearing on or after line 104 needs to be abbreviated as "Z. Marina".
  5. Line 72 and 97, The use of the words 'Blue Carbon' and 'blue carbon' is inconsistent, please express them uniformly.
  6. 1, Please provide longitude and latitude, or add a ruler.
  7. Lines 129-131, Did the sampling time (March, May, July 2019) cover the key developmental stages of seagrass? Additional criteria for defining seedling age (such as whether based on germination time or morphological indicators) are needed. Also, please confirm if Table 1 contains any information about sampling.
  8. Line 159-160, Changed “Reverse 806RB = 5’- GGACTACNVGGGTWTCTAAT-3; (Walters et al. 2015). ” to “Reverse 806RB 5’- GGACTACNVGGGTWTCTAAT-3 (Walters et al. 2015). ”.
  9. Lines 172-173, deleted the sentence “or amplicon sequence variants”.
  10. Lines 200-202, There is no need to separate “Data Accessibility” as a separate Section, it can be placed in the Materials and Methods Section.
  11. Results, It is best to add a few subheadings to describe the Results Section.
  12. Figure 2, The handwriting in this picture is too blurry, please provide a clearer image.
  13. Lines 284, 297, The use of the words 'Nitrogen fixing bacteria' and 'N Fixing Bacteria' is inconsistent, please express them uniformly.
  14. Line 294, deleted “Tarquino et al. 2019)”.
  15. Line 318, Changed “Sulfate Reducing Bacteria” to “Sulfate Reducing Bacteria (SRB)”.
  16. Line 32, Changed “Sulfate Reducing Bacteria” to “SRB”.
  17. Line 445, “Zostera Marina” needs to be italicized.
  18. References 10, 14, These two references are incomplete and need to be supplemented with the page numbers, or article numbers.

Author Response

REVIEWER 1

  1. This work is dedicated to the study of the seagrass microbiome at different stages of plant development, as well as the shifts in functional groups of bacteria that occur during plant aging.

OUR RESPONSE: Thanks

  1. The work was carried out using standard methods, except for the annotation of functional groups within the bacterial community. This approach to annotating potential functional groups is questionable. There are specialized tools available for this purpose, such as Tax4Fun, PanFP, and PicRUST. Why were these not used? By the way, I didn't find table 2.

OUR RESPONSE: We have used one of these specialised tools but verified it against literature. We used FAPROTAX

OUR RESPONSE: Revised table labels in response.

  1. Find and fix misprint.

OUR RESPONSE: done

  1. L 71-73. What are Blue Carbon and Greenhouse Gas Emissions? Are these terms used to describe specific public initiatives? If so, can you provide some links or an explanation of the concepts so that readers don't need to search for this information on their own?

OUR RESPONSE: We have defined blue carbon with link but I think that Greenhouse Gas Emissions shouldn’t need a definition.

  1. L 156. Bacterial community ?

OUR RESPONSE: done

  1. L 208. Sequence annotation has not reached the genus level? Why?

OUR RESPONSE: Many of the sequences were annotated at the Genus level but we chose to focus and report at the Order or Class level as many of the functional classifications of Bacterial taxa at these broader levels are well knon.

  1. L 214. Do you mean Bacteria as Kingdom (taxonomic level) here? Or "bacterial taxa"? Check this.

OUR RESPONSE: Changed to Bacterial taxa

  1. 6 and 7. What is the difference between them? Leave only Fig.6.

OUR RESPONSE: we disagree and feel they both add a visual element to the analysis.

  1. L 294. Find and fix misprint.

OUR RESPONSE: done

  1. Table 1 needs to be formatted.  It's hard to perceive in its modern form.

OUR RESPONSE: done

  1. Figure 2. Here you need to increase the image resolution

OUR RESPONSE: done

Reviewer 2 Report

Comments and Suggestions for Authors

This work is dedicated to the study of the seagrass microbiome at different stages of plant development, as well as the shifts in functional groups of bacteria that occur during plant aging.

The work was carried out using standard methods, except for the annotation of functional groups within the bacterial community. This approach to annotating potential functional groups is questionable. There are specialized tools available for this purpose, such as Tax4Fun, PanFP, and PicRUST. Why were these not used? By the way, I didn't find table 2.

L51. Find and fix misprint.

L 71-73. What are Blue Carbon and Greenhouse Gas Emissions? Are these terms used to describe specific public initiatives? If so, can you provide some links or an explanation of the concepts so that readers don't need to search for this information on their own?

L 156. Bacterial community ?

L 208. Sequence annotation has not reached the genus level? Why?

L 214. Do you mean Bacteria as Kingdom (taxonomic level) here? Or "bacterial taxa"? Check this.

Fig.6 and 7. What is the difference between them? Leave only Fig.6.

L 294. Find and fix misprint.

Table 1 needs to be formatted.  It's hard to perceive in its modern form.

Figure 2. Here you need to increase the image resolution

Author Response

REVIEWER 2

this manuscript systematically investigated the functional differences of rhizosphere and endophytic microbial communities in seagrass (Zostera Marina) at different developmental stages (seedlings and adults), revealing the dynamic role of microbial communities in sulfur environment adaptation, nitrogen sulfur cycling, and blue carbon storage. The research results provide a theoretical basis for microbial community regulation in seagrass restoration, with clear ecological significance and application potential. Overall, this manuscript merits publication after major revision. The specific comments are presented below.

  1. The scope of the title of this manuscript is a bit broad, it is suggested to add Zostera marina to the title.

OUR RESPONSE: Changed

  1. Abstract, Please confirm if “seeding” and “judiciary” have the same meaning and if they need to be expressed uniformly.

OUR RESPONSEI apologise here but we don’t understand the meaning of this comment.

  1. Lines 41-42, Generally speaking, only the species names of microorganisms need to be italicized, while other names such as phyla, classes, orders, and families do not need to be italicized. Therefore, the words "Gamma and Deltaproteobacteria" in this sentence do not need to be italicized.

OUR RESPONSE: Changed

  1. Line 51, Z. marina appears for the first time in the main text and requires the full name "Zostera marina" (italicized), while "Zostera Marina" appearing on or after line 104 needs to be abbreviated as "Z. Marina".

OUR RESPONSE: Changed

  1. Line 72 and 97, The use of the words 'Blue Carbon' and 'blue carbon' is inconsistent, please express them uniformly.

OUR RESPONSE: Changed

  1. 1, Please provide longitude and latitude, or add a ruler.

OUR RESPONSE: Lat longs added

  1. Lines 129-131, Did the sampling time (March, May, July 2019) cover the key developmental stages of seagrass? Additional criteria for defining seedling age (such as whether based on germination time or morphological indicators) are needed. Also, please confirm if Table 1 contains any information about sampling.

OUR RESPONSE: We’ve clarified the sampling time making reference to another paper conducted at the site. We have removed reference to table 1 as this doesn’t refer to sampling.

  1. Line 159-160, Changed “Reverse 806RB = 5’- GGACTACNVGGGTWTCTAAT-3; (Walters et al. 2015). ” to “Reverse 806RB 5’- GGACTACNVGGGTWTCTAAT-3 (Walters et al. 2015). ”.

OUR RESPONSE: done

  1. Lines 172-173, deleted the sentence “or amplicon sequence variants”.

OUR RESPONSE: Done

  1. Lines 200-202, There is no need to separate “Data Accessibility” as a separate Section, it can be placed in the Materials and Methods Section.

OUR RESPONSE: done

  1. Results, It is best to add a few subheadings to describe the Results Section.

OUR RESPONSE: The way we’ve written the results we don’t feel sections are helpful.

  1. Figure 2, The handwriting in this picture is too blurry, please provide a clearer image.

OUR RESPONSE: Done

  1. Lines 284, 297, The use of the words 'Nitrogen fixing bacteria' and 'N Fixing Bacteria' is inconsistent, please express them uniformly.

OUR RESPONSE: Done

  1. Line 294, deleted “Tarquino et al. 2019)”.

OUR RESPONSE: done

  1. Line 318, Changed “Sulfate Reducing Bacteria” to “Sulfate Reducing Bacteria (SRB)”.

OUR RESPONSE: done

  1. Line 32, Changed “Sulfate Reducing Bacteria” to “SRB”.

OUR RESPONSE: Done

  1. Line 445, “Zostera Marina” needs to be italicized.

OUR RESPONSE: done

  1. References 10, 14, These two references are incomplete and need to be supplemented with the page numbers, or article numbers.

OUR RESPONSE: done

Reviewer 3 Report

Comments and Suggestions for Authors

Dear authors,

The manuscript is interesting, with many important and relevant data, but at the same time it presents a series of weaknesses that require the authors to enrich its content, according to the observations below. Even though molecular biology techniques are extremely useful, in some cases, they alone only suggest the presence of a microbial group and some processes in natural environments, not their real dynamics. Therefore, along with molecular biology, a series of physicochemical data from the investigated habitats are necessary. In this sense, it is well known that some groups and species may be inactive although viable, not having a significant impact on the transformations in the studied habitats. These microorganisms can, however, be detected by molecular biology techniques. Therefore, we consider that it would be more useful to supplement the information related to the major microbial processes, with more details, at least based on the literature.

Discussion section. Therefore, we believe that the authors should further detail the functional role of the respective microbial groups and species and their specific connections with plants in their different stages of development. The holobiont involves an integration of the activities of the two partners: microbiome and plants. The manuscript does not clearly state the particularities and specificity of the interaction between microorganisms and Zostera marina. What are the particularities of the interaction between this plant species and microorganisms? Are there differences in the composition of the microbiome compared to other marine plant-microorganism holobionts? Also, the article lacks a series of relevant physicochemical analyses of the natural niches represented by the plant, its rhizosphere, and sediments devoid of plant influence.

Regarding the simultaneous presence of nitrifying and sulfate-reducing bacteria - the first group is aerobic while the second group is anaerobic. How do the authors explain the presence and activity of these two groups and their interaction with Zostera?

Conclusions section. I believe that authors should bring the original results to the fore in comparison with other more or less similar studies. This would lead to a better understanding of the results presented in the article, as well as to increased interest from the readers.

Best regards!

Author Response

The manuscript is interesting, with many important and relevant data, but at the same time it presents a series of weaknesses that require the authors to enrich its content, according to the observations below. Even though molecular biology techniques are extremely useful, in some cases, they alone only suggest the presence of a microbial group and some processes in natural environments, not their real dynamics. Therefore, along with molecular biology, a series of physicochemical data from the investigated habitats are necessary. In this sense, it is well known that some groups and species may be inactive although viable, not having a significant impact on the transformations in the studied habitats. These microorganisms can, however, be detected by molecular biology techniques. Therefore, we consider that it would be more useful to supplement the information related to the major microbial processes, with more details, at least based on the literature.

Discussion section. Therefore, we believe that the authors should further detail the functional role of the respective microbial groups and species and their specific connections with plants in their different stages of development.

OUR RESPONSE: We have added discussion

The holobiont involves an integration of the activities of the two partners: microbiome and plants. The manuscript does not clearly state the particularities and specificity of the interaction between microorganisms and Zostera marina. What are the particularities of the interaction between this plant species and microorganisms? Are there differences in the composition of the microbiome compared to other marine plant-microorganism holobionts? Also, the article lacks a series of relevant physicochemical analyses of the natural niches represented by the plant, its rhizosphere, and sediments devoid of plant influence.

OUR RESPONSE: We have added a table to include these parameters.

Regarding the simultaneous presence of nitrifying and sulfate-reducing bacteria - the first group is aerobic while the second group is anaerobic. How do the authors explain the presence and activity of these two groups and their interaction with Zostera?

OUR RESPONSE: Seagrasses are well know to be able to modify their rhizosphere environment via Radical oxygen loss (ROL) (Jensen et al., 2007; Martin et al. 2019), this leads to the presence of oxygen in the Rhizosphere that can support Nitrifying bacteria. Additionally recent work has shown that the Seagrass Posidonia oceanica has an endophytic nitrifying bacteria within the roots . As our study did not investigate this directly for Zostera marina we were able to extract DNA directly from the seagrass roots, therefore it could be investigated further. This is discussed in paragraph 318-334

Jensen, S.I., Kühl, M., and Priemé, A. (2007). Different bacterial communities associated with the roots and bulk sediment of the seagrass Zostera marina. FEMS Microbiology Ecology 62, 108-117.

Martin, B.C., Bougoure, J., Ryan, M.H., Bennett, W.W., Colmer, T.D., Joyce, N.K., Olsen, Y.S. and Kendrick, G.A., 2019. Oxygen loss from seagrass roots coincides with colonisation of sulphide-oxidising cable bacteria and reduces sulphide stress. The ISME journal, 13(3), pp.707-719.

Mohr, W., Lehnen, N., Ahmerkamp, S., Marchant, H.K., Graf, J.S., Tschitschko, B., Yilmaz, P., Littmann, S., Gruber-Vodicka, H., Leisch, N. and Weber, M., 2021. Terrestrial-type nitrogen-fixing symbiosis between seagrass and a marine bacterium. Nature600(7887), pp.105-109.

Conclusions section. I believe that authors should bring the original results to the fore in comparison with other more or less similar studies. This would lead to a better understanding of the results presented in the article, as well as to increased interest from the readers.

OUR RESPONSE: We have added literature to this effect.

Round 2

Reviewer 1 Report

Comments and Suggestions for Authors

The author has already responded to the issues I am concerned about one by one, and I have no further comments.